# Role of Allogeneic Hematopoietic Stem Cell Transplantation for Philadelphia Chromosome-Positive B-Cell Acute Lymphoblastic Leukemia in the Contemporary Era

**DOI:** 10.3390/cancers17010104

**Published:** 2024-12-31

**Authors:** Omer Jamy, Talha Badar

**Affiliations:** 1Division of Hematology and Oncology, Department of Medicine, University of Alabama at Birmingham, 1720 2nd Avenue S, NP2540W, Birmingham, AL 35294, USA; 2Division of Hematology and Oncology, Department of Medicine, Mayo Clinic, Jacksonville, FL 32224, USA; badar.talha@mayo.edu

**Keywords:** acute lymphoblastic leukemia, Ph+, allogeneic transplant

## Abstract

Allogeneic hematopoietic stem cell transplantation (allo-HSCT) has always been an integral part of the treatment algorithm of Philadelphia chromosome-positive B-cell acute lymphoblastic leukemia (Ph+ ALL). Recently, the approval of novel therapies such as blinatumomab, inotuzumab ozogamicin and chimeric antigen receptor T-cell (CAR-T) therapy have improved outcomes of B-cell ALL. With potent kinase inhibitors and novel targeted therapy, the treatment guidelines for Ph+ ALL are evolving rapidly. With all these advances, the role of allo-HSCT in the management of Ph+ ALL needs to be re-examined. Herein, we discuss the current evidence for the utilization of allo-HSCT for Ph+ ALL.

## 1. Introduction

Philadelphia chromosome-positive B-cell acute lymphoblastic leukemia (Ph+ B-cell ALL) is characterized by the presence of t(9;22), resulting in *BCR::ABL1* gene rearrangement. Its incidence increases with age, accounting for nearly half of ALL cases in patients ≥ 60 years old [1,2,3]. Historically, the treatment of Ph+ ALL consisted of intensive induction chemotherapy followed by allogeneic hematopoietic stem cell transplantation (allo-HSCT). However, long-term outcomes with this approach were suboptimal, with 5-year survival ranging from 10% to 30% [4]. The treatment landscape of Ph+ ALL has been reshaped over the past two decades with the introduction of *BCR::ABL1* tyrosine kinase inhibitors (TKIs), resulting in dramatic improvements in long-term survival [1,2]. Immune-based therapies such as blinatumomab (CD3-CD19 bispecific antibody), inotuzumab ozogamicin (anti-CD22 antibody drug conjugate) and chimeric antigen receptor T-cell (CAR-T) therapy in relapse and refractory (R/R) ALL have further improved the outcomes of B-cell ALL, including Ph+ ALL [5,6,7,8,9,10,11]. With potent TKIs and novel targeted therapy, we are in the midst of a paradigm shift in the treatment strategy for Ph+ ALL, potentially moving towards low-intensity or chemotherapy-free regimens. Additionally, with improved tools for detecting measurable residual disease (MRD), a risk-adapted approach for the treatment of Ph+ ALL is becoming more acceptable. With all these advances, the role of allo-HSCT in the management of Ph+ ALL needs to be re-examined. Herein, we discuss the current evidence for the utilization of allo-HSCT for Ph+ ALL, focusing on novel therapies and MRD-directed care.

## 2. Current Standard

The current standard of care for patients with Ph+ ALL is a *BCR::ABL1* TKI combined with chemotherapy [12,13]. The addition of imatinib, a first-generation TKI, to intensive chemotherapy for newly diagnosed Ph+ ALL resulted in a higher complete remission (CR) rate (92% vs. 82%); however, long-term survival (4-year overall survival [OS] rate; 38% vs. 22%) was modestly improved due to the high rate of relapse [14]. The introduction of second-generation TKIs such as dasatinib and nilotinib further improved outcomes (5-year OS 35–45%) for these patients, with decreased incidences of relapse [15,16,17]. The combination of dasatinib and hyperCVAD (hyperfractionated cyclophosphamide, vincristine, doxorubicin, and dexamethasone, alternating with high-dose methotrexate and cytarabine) for newly diagnosed Ph+ ALL resulted in a 65% complete molecular remission (CMR; *BCR::ABL1* ≤ 0.01%) rate and a 5-year OS of 46% [16]. The addition of nilotinib to intensive chemotherapy resulted in a cumulative CMR rate of 83% and a 4-year OS of 45% [15]. Another randomized trial of nilotinib, with or without cytarabine during consolidation, showed no difference in major molecular response (71% vs. 77%) or 4-year survival (79% vs. 73%). However, an increased risk of relapse was observed where cytarabine was omitted (31%) vs. not omitted (13%) [18]. Despite the use of first- and second-generation TKIs to improve survival, the long-term outcomes remained modest. One of main reasons is the emergence of single-mutation variants including T315I that are resistant to first- and second-generation TKIs [1,2]. The T315I mutation can occur in up to 75% of patients with Ph+ ALL at the time of relapse when treated with earlier-generation TKIs [19]. Ponatinib, a third-generation TKI with activity against T315I mutation, has shown promising results in combination with both chemotherapy and immunotherapy for newly diagnosed Ph+ ALL [20,21]. Propensity-matched analyses have suggested that compared to both first- and second-generation TKIs, ponatinib may lead to improved long-term outcomes due to both deeper responses, as well as the suppression of resistant variants, including T315I [22]. More recently, the PhALLCON study, a randomized phase III clinical trial of imatinib vs. ponatinib in combination with low-intensity chemotherapy in newly diagnosed Ph+ ALL, demonstrated a higher rate of MRD-negative CR at 3 months with ponatinib (34.4%) compared to imatinib (16.7%), meeting the primary endpoint of the study (*p* = 0.002). The study led to the approval of ponatinib in newly diagnosed Ph+ ALL [23]. There is currently no prospective data available to select between second-generation TKIs and ponatinib as the treatment of choice in Ph+ ALL.

In the current era, most patients with Ph+ ALL will either receive a second- or third-generation TKI as part of induction therapy. While the current standard is to combine a TKI with chemotherapy, single-arm studies of blinatumomab and TKI have shown promising results as well. The combination of dasatinib and blinatumomab as upfront therapy resulted in a CMR rate of 60% with half the patients proceeding to allo-HSCT. The 3-year OS rate for the cohort was 80%. There were four isolated central nervous system (CNS) relapses out of the nine patients who relapsed, highlighting the need for CNS-directed therapy with novel agents. The emergence of the T315I mutation and the presence of the *IKZF1* mutation and additional molecular abnormalities were associated with poor outcomes with the combination [24]. In a SWOG study of older patients (median age: 73 years [range: 65–87 years]), the combination of dasatinib and blinatumomab resulted in a 3-year OS rate of 87% [25]. Subsequently, a phase II trial of ponatinib and blinatumomab was conducted, enrolling 30 patients and resulting in an overall CMR rate of 86% with an estimated 2-year OS of 93%. Only one patient proceeded to allo-HSCT on this trial. Longer follow-up will facilitate data interpretation but for the time being these results appear promising [20]. An ongoing phase III study by the ECOG-ACRIN cooperative group (EA9181; NCT04530565) is comparing TKI–blinatumomab to TKI and hyperCVAD (TKI choice includes dasatinib or ponatinib) as frontline therapy for Ph+ ALL. Another ongoing phase III trial (GIMEMA; NCT04722848) is comparing ponatinib–blinatumomab to imatinib–chemotherapy in newly diagnosed Ph+ ALL.

Before the discovery of TKIs, Ph+ ALL was considered to have dismal outcomes. With the use of imatinib and allo-HSCT, long-term outcomes saw a significant improvement with a 5-year OS rate approaching 50% [26,27]. Novel TKIs and immunotherapy combinations have improved these outcomes even further. Understanding the biology of Ph+ ALL and associating it with treatment and outcomes can help to identify high-risk and low-risk patients within the disease. Several studies have found genetic anomalies, such as deletions of IKZF1, CDKN2A/2B and PAX5, to adversely affect outcomes in Ph+ ALL, even when treated with dasatinib and blinatumomab [2,24,28]. The presence of IKZF1 (plus) abnormalities was also associated with inferior outcomes when treated with ponatinib and hyperCVAD [2,22]. Longer follow-up of the ponatinib and blinatumomab cohort is needed to see if the combination is able to overcome the adverse outcomes of IKZF1 (plus) abnormalities.

## 3. Allogeneic Hematopoietic Stem Cell Transplantation

### 3.1. Transplant in Pre-TKI Era vs. 1st-Generation TKI vs. 2nd/3rd-Generation TKI Combinations

In the pre-imatinib era, the UKALLXII/ECOG 2993 trial evaluated the role of allo-HSCT in the management of Ph+ ALL in younger patients. In 267 patients, with a median age of 40 years, 28% proceeded to allo-HSCT in first remission. The 5-year OS rates for matched-sibling allo-HSCT, matched-unrelated allo-HSCT and chemotherapy only were 44%, 36% and 19%, respectively. In an adjusted analysis, only relapse-free survival (RFS) was found to be superior with allo-HSCT. Transplant-related mortality (TRM) was 27% in sibling allo-HSCT and 39% in unrelated allo-HSCT, significantly higher than what would be expected in the current era. All patients received myeloablative conditioning (MAC), which predisposed them to an increased risk of TRM. Furthermore, the deaths in remission (TRM) were higher in the presence of GVHD compared to the absence of GVHD, signifying GVHD as one of the main contributors to TRM. As this study was conducted from 1993 to 2004, the TRM rates are not entirely surprising. However, due to these high rates, the trial did not demonstrate a survival benefit in favor of allo-HSCT after adjustment, and the main advantage was observed in RFS only. Nonetheless, this trial was able to provide a pivotal baseline against which future developments could be compared [14].

In the GIMEMA LAL 0904 trial, patients with Ph+ ALL were treated with imatinib and chemotherapy with or without allo-HSCT. There were 51 patients in the study with a median age of 45.9 years (range 16.9–59.7 years). Out of these 51 patients, 44 were eligible for allo-HSCT per protocol and 20 proceeded to transplant. Reasons for not proceeding to allo-HSCT included medical decision (*n* = 8), relapse (*n* = 7), toxicity (*n* = 5), no donor (*n* = 3) and patient refusal (*n* = 1). As a time-dependent covariate, allo-HSCT led to superior disease-free survival (DFS) and OS in the trial [29].

The role of allo-HSCT was also investigated in the context of second-generation TKIs. A phase II trial of younger patients (<60 years) with Ph+ ALL evaluated the combination of dasatinib plus hyperCVAD, followed by allo-HSCT. Post-transplant maintenance was permitted during the trial. The primary endpoint of the trial was 12-month RFS after transplant. In 94 evaluable patients with a median age of 44-years (range 20–60 year), 88% achieved remission and 41 patients proceeded to allo-HSCT in CR1. The 3-year OS rate for the entire cohort was 69%. For those proceeding to transplant, the 12-month RFS was 83%, significantly higher than the historic rate of 40%. The 3-year RFS post-transplant was 76%. In a landmark analysis (175 days after achieving remission), allo-HSCT was associated with superior RFS (*p* = 0.038) and OS (*p* = 0.037). In Cox regression models with allo-HSCT as a time-dependent covariate and adjustments for age, white blood cell (WBC) count and prior therapy, transplant was associated with superior OS (*p* = 0.037) [17].

The combination of ponatinib and hyperCVAD was investigated in a single-arm phase II trial of 86 patients with Ph+ ALL. Sixty-six patients were newly diagnosed and twenty received one or two cycles of prior therapy with dasatinib and hyperCVAD. Overall, 20 patients proceeded to allo-HSCT in first remission. Sixteen of these patients had a transcript of ≤0.1% and four had a level of >0.1% at the time of transplant. Thirteen patients continued with a TKI as post-transplant maintenance (ponatinib = 7; dasatinib = 3; imatinib = 2; nilotinib = 1). Five patients had non-relapse mortality and one died from relapsed disease. The median time to transplant from the start of therapy was 8 months. An eight-month landmark analysis did not show a difference in survival between patients proceeding to transplant versus those who did not. The 6-year OS was 70% and 87% for the transplant and non-transplant groups, respectively (*p* = 0.13) [30]. In the PONALFIL trial, ponatinib was combined with daunorubicin, vincristine and prednisone with 26 patients proceeding to allo-HSCT. The majority of the patients received MAC and the 3-year event-free survival (EFS) and OS for these patients were 70% and 96%, respectively. One patient died from TRM after 3 years [31].

### 3.2. Transplant After Upfront Immunotherapy Plus TKIs

Most recently, in the D-ALBA trial (dasatinib plus blinatumomab) discussed previously, 30 out of 63 patients proceed to allo-HSCT as consolidation (6 in second remission). Out of the 30 patients, 23 were not in molecular remission at the time of transplant. The TRM was 10%. Although the sample size was small along with heterogenous remission status, allo-HSCT did not impact DFS or OS during multivariable analysis [32].

In the study of ponatinib plus blinatumomab, only two out of 60 patients proceeded to allo-HSCT in first remission. With a median follow-up of 24 months, the 3-year EFS and OS were 77% and 91%, respectively. Seven patients relapsed (2 = systemic relapse; 4 = isolated CNS relapse; 1 = extramedullary Ph-negative ALL relapse) [20,33].

In the intermediate analysis of the GIMEMA ALL2820 trial, 133 patients received ponatinib plus blinatumomab. The decision to proceed to allo-HSCT was based on IKZF1 (plus) at diagnosis and/or MRD persistence. To date, 12% of the patients have proceeded to transplant, and with a median follow-up of 6.4 months, the estimated 18-month OS is 91.6% [34].

The post-transplant outcomes of Ph+ ALL in the selected clinical trials are highlighted in Table 1.

### 3.3. Transplant Based on the Depth of Response/Measurable Residual Disease (MRD)

The role of MRD in ALL is well defined, with the absence of MRD being associated with superior outcomes [35]. For Ph+ ALL, the reverse transcription polymerase chain reaction (RT-PCR) for BCR::ABL1 has historically been used to determine MRD. More recently, studies have suggested that RT-PCR for BCR::ABL1 may not be the best measure of MRD in several cases of Ph+ ALL and that a high sensitivity next-generation sequencing (NGS)-based immunosequencing assay (clonoSEQ) could potentially offer a more accurate assessment of prognosis [36,37]. Short et al. demonstrated 32% discordance in MRD assessment by RT-PCR and NGS. In their study, patients who were MRD-negative by NGS but positive by RT-PCR (PCR+/NGS−) did not benefit from therapeutic intervention with the persistence of stable PCR levels. Interestingly, none of those patients relapsed. These findings contrasted with patients who had both NGS and PCR positivity (PCR+/NGS+), where PCR levels responded to therapeutic intervention. In a comparison of patients with PCR+/NGS− with those who were PCR-/NGS-, the relapse-free survival and OS rates were similar, suggesting that PCR for BCR::ABL1 did not add to the prognostic information for patients who achieved NGS-based MRD negativity [36].

In the era of TKIs, the quantitative monitoring of MRD kinetics during therapy has aided predicting outcomes of patients with Ph+ ALL. In a study focusing on outcomes of allo-HSCT in patients with Ph+ ALL treated with imatinib-based chemotherapy, MRD kinetics using PCR-based testing was able to identify subgroups at the highest risk of relapse. The study divided 95 patients into 4 response groups based on MRD kinetics after 2 courses of imatinib plus chemotherapy: (1) early responders (patients with MMR/CMR4.5 after 2 courses), (2) late responders (patients with conversion from no MMR to MMR/CMR4.5 after 2 courses), (3) intermediate responders (patients with MRD levels of >0.1 to 1% after 2 courses) and (4) poor responders (patients with MRD levels of >1% after 2 courses). In multivariable analysis, the strongest predictor for long-term post-transplant outcomes was MRD kinetics. Patients with early responses had the lowest risk of relapse compared to those with intermediate and poor responses. Similar results were observed for DFS outcomes. Interestingly, MRD after one cycle of therapy was not significant in multivariable analysis, highlighting that the kinetics of MRD are more closely associated with outcomes [38].

Along these lines, Short et al. evaluated 85 patients with Ph+ ALL to investigate the impact of achieving CMR on survival outcomes. None of the 85 patients proceeded to allo-HSCT. At the 3-month time point, 51 patients were in CMR, 16 were in MMR and 18 were less than MMR. They reported that MRD status at 3 months was a better predictor of OS and RFS compared to MRD status at the time of achieving complete remission. Furthermore, achieving MRD negative status at 3 months lead to improved OS and RFS compared to not achieving CMR at 3 months. These results suggested that patients with Ph+ ALL achieving CMR at 3 months have excellent long-term outcomes and may be able to avoid allo-HSCT in first remission [39].

Achieving CMR in Ph+ ALL is now recognized as a powerful prognostic factor. As MRD kinetics might play a more important role in the risk stratification of this population, a couple of recent studies have investigated the role of allo-HSCT in this context. In the first study, Ghobadi et al. focused only on those patients who achieved CMR by 90 days after diagnosis and questioned the benefit of allo-HSCT in this population [40]. The multi-institutional retrospective study included 230 patients with Ph+ ALL in CMR at day 90 after diagnosis (allo-HSCT = 98, no allo-HSCT = 132). More patients in the non-transplant group received ponatinib (29% vs. 6%), whereas dasatinib (54% vs. 47%) and imatinib (40% vs. 24%) use was more common in the transplant group. To account for baseline differences in characteristics, the authors performed a multivariable analysis, as well as a propensity score-matched analysis, to understand the results. In the multivariable analysis, after adjusting for age, performance status and the use of ponatinib as upfront therapy, allo-HSCT was not associated with improvements in OS or RFS. The cumulative incidence of relapse (CIR) was significantly lower in patients who underwent allo-HSCT. These survival results were then confirmed in the propensity score matched analysis as well. As non-relapse mortality (NRM) is a major cause of transplant failure, a sub-analysis was performed focusing only on reduced-intensity conditioning (RIC) and again no difference was observed in survival outcomes. Noteworthy limitations of the study include its retrospective nature, along with the time period that it analyzed (2001–2018). Given that the transplant platform has evolved significantly in the past few years in terms of donor types, graft versus host disease (GVHD) prevention and infection control, the high NRM observed in this study might differ from what is considered standard today. Lastly, acknowledging that the practice is very heterogenous, TKI-maintenance post-transplant was implemented in less than half the patients in the study. Nonetheless, we have compiled evidence here suggesting that the role of allo-HSCT could be limited in a subset of patients with Ph+ ALL and that future research utilizing modern risk stratification tools may help to resolve unanswered questions.

In another large real-world, multi-institutional analysis evaluating the impact of induction intensity, as well as allo-HSCT, on the survival of patients with Ph+ ALL, the authors noted that those who achieved CMR at the 3-month time-point had improved RFS if they underwent allo-HSCT in first remission [41]. Similar to the previous study, allo-HSCT did not add an OS benefit for patients in CMR at 3 months. For the overall population, allo-HSCT was associated with improved RFS and OS in multivariable- and propensity score-matched analysis. Within the limitations of any real-world study, these data add to evidence suggesting that perhaps not all patients with Ph+ ALL need to proceed to allo-HSCT in first remission.

### 3.4. Recent Advances in the Transplant Platform

Although allo-HSCT has historically been considered a high-risk procedure with morbidity and mortality related to infectious complications and GVHD, we have seen significant improvements in the platform over the past few years [42,43,44,45]. As the median age of patients diagnosed with a high-risk myeloid malignancy is typically in the late 60s or early 70s, several studies have shown the feasibility of allo-HSCT in the elderly population [46]. One of the biggest developments of the past decade was the ability to offer haploidentical-HSCT (haplo-HSCT) to patients who lacked a fully matched-related or -unrelated donor [47]. Through haplo-HSCT, we have been able to expand the pool of patients to whom we can offer transplant as a treatment option. The success of haplo-HSCT has relied on the ability of post-transplant cyclophosphamide (PTCy) to mitigate the risk of GVHD. Recent registry studies have evaluated the outcomes of patients with ALL treated with haplo-HSCT to better define the role of this platform. In a Center for International Blood and Marrow Transplant Research (CIBMTR) analysis comparing haplo-HSCT to matched-related, matched-unrelated and cord blood transplant in both Ph+ and Ph− ALL patients, the authors noted that there was no difference in OS between haplo-HSCT and fully matched-related or -unrelated donors. In fact, the risk of chronic GVHD was lower in patients undergoing haplo-HSCT. Compared to 7/8 matched-related or cord blood transplant patients, haplo-HSCT was associated with improved OS in patients with ALL. These results point to the feasibility of haplo-HSCT in ALL, and future studies will help with establishing its role in the treatment algorithm for Ph+ ALL [45]. Given the remarkable success of GVHD prevention with PTCy in haplo-HSCT, the Blood and Marrow Transplant Clinical Trials Network (BMT CTN) conducted a randomized, multicenter, phase III trial comparing PTCy-based GVHD prophylaxis to standard calcineurin inhibitor-based GVHD prophylaxis in fully matched-related or -unrelated RIC allo-HSCT [42]. The primary end point of the study was 1-year GVHD-free, relapse-free survival (GRFS). The study met its primary endpoint with the 1-year GRFS being 52.7% in the PTCy arm and 34.9% in the control arm, thus making PTCy the new standard for RIC transplants. The incidence of grade III-IV acute GVHD and severe chronic GVHD was significantly reduced in the PTCy arm. As GVHD remains a major cause of transplant failure and NRM, these promising results should enable more patients to undergo allo-HSCT in general. In acute myeloid leukemia (AML) and myelodysplastic syndrome (MDS), MAC is associated with a higher TRM but still leads to improved survival when compared to reduce intensity conditioning (RIC). This benefit is mainly driven by the high rates of relapse-related deaths with RIC [48]. In Ph+ ALL, randomized data comparing MAC vs. RIC in the contemporary era are limited, but several registry studies, as well as large single-center reports, have demonstrated comparable outcomes with either intensity, especially in older patients, as well as those in MRD-negative remission prior to transplant. These studies highlight that RIC is a suitable alternative in patients unable to tolerate MAC [49,50]. To address the issue of post-transplant relapses in Ph+ ALL, the role of TKIs, as maintenance, has been extensively investigated. The majority of the studies demonstrated that maintenance with a TKI for 2 or 3 years post-transplant is associated with a lower risk of relapse, as well as improved progression-free survival [51,52]. In a European registry study, the administration of post-transplant TKI was associated with an overall survival benefit as well [53]. Based on the available data, the current practice across most centers is to offer post-transplant maintenance with a TKI in Ph+ ALL. Other strategies on the horizon include the use of blinatumomab and even CAR-T therapy as maintenance in ALL [54]. However, these studies are early in their conception, and the field eagerly awaits their findings.

## 4. CAR-T Therapy in Ph+ ALL

There are currently three approved CAR-T products for R/R ALL (brexucabtagene autoleucel, tisagenlecleucel and obecabtagene autoleucel). The trials leading to the approval of these products did include a proportion of patients with Ph+ ALL and outcomes were generally favorable. More recently, studies combining TKI with CAR-T in early-line therapy of Ph+ ALL have yielded promising results. In a phase II trial of 18 patients with newly diagnosed Ph+ ALL, dasatinib followed by CD19 and CD22 CAR-T-cell infusions resulted in a complete molecular response in 76.9% of the patients. With a median follow-up of 13.5 months, 16 patients were in complete hematological remission, with 14 of those in complete molecular remission. None of these patients proceeded to allo-HSCT [55]. A recent phase I trial of 13 patients (<65 years) with newly diagnosed Ph+ ALL investigated multicycle-sequential anti-CD19 CAR-T plus autologous CD19+ feeding T cells in combination with a TKI as consolidation therapy. No patients were transplanted, and after a median follow-up of 27 months, overall survival was 83% and relapse-free survival was 84% [56]. Although promising, these early results need confirmation in larger studies to determine if CAR-T therapy can potentially replace allo-HSCT in high-risk Ph+ ALL.

## 5. Relapse and Refractory Disease

The treatment options for R/R ALL include blinatumomab, inotuzumab ozogamicin and CAR-T therapy. For Ph+ ALL, novel TKIs may be added to these agents. The potential benefits of blinatumomab and CAR-T in Ph+ ALL have been extensively discussed before. Studies have shown that inotuzumab ozogamicin, with or without a TKI, is feasible to administer in R/R Ph+ ALL. It can serve as a potential bridge to allo-HSCT in second remission, although more data are needed to understand the potential post-transplant toxicities of using it pre-transplant. The studies so far do not show an increased risk of veno-occlusive disease. However, these studies had small sample sizes and larger studies are needed to assess the full benefits, as well as the safety, of inotuzumab ozogamicin in both newly diagnosed and R/R Ph+ ALL [57,58]. Additionally, novel TKIs, such as olverembatinib, have shown encouraging single-agent activity in R/R Ph+ ALL and are now being investigated in various combinations with either chemotherapy or blinatumomab [59]. As these agents are moved to the frontline setting with the hope of inducing deeper and durable remissions, a challenge may arise with regard to treating those patients with disease relapse after these therapies. After thorough testing to characterize the nature of the relapse disease in terms of antigen expression and molecular anomalies, treatment may largely depend on the ability to enroll on clinical trials or the utilization of standard-of-care TKIs, immunotherapy or even conventional chemotherapy.

## 6. Conclusions

The treatment paradigm of Ph+ ALL has evolved considerably over the past two decades. The introduction of first-generation TKIs, combined with chemotherapy, established TKIs as the backbone of the treatment of Ph+ ALL. With second- and third-generation TKIs, outcomes only continue to improve. Although chemotherapy still has a role in the treatment of ALL, the results with novel agents such as blinatumomab, inotuzumab ozogamicin and CAR-T have been impressive, and these agents are being actively investigated in the frontline setting. In Ph+ ALL, chemotherapy-free regimens of blinatumomab plus either second- or third-generation TKIs have shown excellent results in the frontline setting, and ongoing randomized trials will inform us whether conventional chemotherapy still has a role in the management of this disease. Allo-HSCT has always been an integral part of the treatment algorithm of Ph+ ALL, with cure being the goal. As the transplant platform continues to become safer with respect to NRM, its applicability will broaden, especially for older patients. Nonetheless, it still carries a risk of morbidity and mortality and if an opportunity arises to re-evaluate its role for a specific disease, it should be pursued. In the current era, all transplant eligible patients with Ph+ ALL should have a consultation with a transplant expert to discuss the risks and benefits of proceeding with allo-HSCT. Based on the evidence presented in this review, all eligible patients with Ph+ ALL in second remission, those who do not achieve a CMR after three months of frontline therapy and patients harboring high-risk biological anomalies such as IKZF1 should be advised to undergo allo-HSCT. There are observational data that suggest that those patients who can achieve CMR at three months do not obtain a survival advantage by proceeding to transplant (Figure 1). However, allo-HSCT does decrease the risk of relapse for this population and, therefore, a shared-decision making strategy involving the patient, the transplant physician and the leukemia physician is required. The results of ongoing trials, especially the combination of immunotherapy with TKIs in the frontline setting, will shed more light on the role of chemotherapy, as well as allo-HSCT, in the treatment of patients with Ph+ ALL.

## Figures and Tables

**Figure 1 cancers-17-00104-f001:**
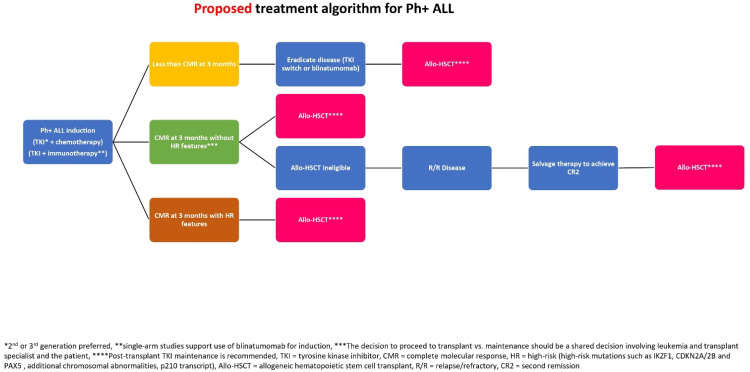
Proposed treatment algorithm for Ph+ ALL.

**Table 1 cancers-17-00104-t001:** Post-transplant outcomes in selected Ph+ ALL clinical trials.

Trial	Therapy	Allo-HSCT Patients (*n*)	Donor Type	Conditioning	Outcomes	TRM
UKALLXII/ECOG 2993 (PMID: 24277073)	DNR/VCR/PRED/L-ASP	75	MRD/MUD	MAC	5-year OSMRD: 44% MUD: 39%	MRD: 27% MUD: 39%
GIMEMA LAL 0904 (PMID: 27515250)	Imatinib + HAM	20	MRD/MUD/Haplo	MAC	OS and DFS benefit with allo-HSCT	-
US intergroup (PMID: 29046900)	Dasatinib + HyperCVAD	41	MRD/MUD	MAC	12 m RFS: 83% 12 m OS: 87%	-
MDACC (PMID: 36600670)	Ponatinib + HyperCVAD	20	MRD/MUD/Haplo	-	6-year OS: 70%	5/20 (6 years)
D-ALBA (PMID: 33085860)	Dasatinib + Blinatumomab	30	MRD/MUD/Haplo	-	23 patients alive at median follow-up of 49 m	13.7% (1 year)
PONALFIL (PMID: 35675590)	Ponatinib + DNR/VCR/PRED	26	MRD/MUD	Mainly MAC	3-year EFS: 70% 3-year OS: 96%	1/26 (3 years)
JALSG (PMID: 38314662)	Imatinib or Dasatinib + DRN/VCR/PRED/CPM	101	MRD/MUD	Mainly MAC	5-year RFS: 70% 5-year OS: 73%	17% (5 years)

Allo-HSCT = allogeneic hematopoietic stem cell transplantation; TRM = transplant-related mortality; DNR = daunorubicin; VCR = vincristine; PRED = prednisone; L-ASP = L-asparaginase; HAM = cytarabine, mitoxantrone and G-CSF; HyperCVAD = hyperfractionated cyclophosphamide (CPM), vincristine, doxorubicin, and dexamethasone, alternating with high-dose methotrexate and cytarabine; MRD = matched-related donor; MUD = matched-unrelated donor; haplo = haploidentical donor; MAC = myeloablative conditioning; OS = overall survival; DFS = disease-free survival; RFS = relapse-free survival; EFS = event-free survival.

## Data Availability

Not applicable.

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
