# Peer review of "Role of Allogeneic Hematopoietic Stem Cell Transplantation for Philadelphia Chromosome-Positive B-Cell Acute Lymphoblastic Leukemia in the Contemporary Era"

_cancers, 2024, doi:10.3390/cancers17010104_

Round 1

Reviewer 1 Report

Comments and Suggestions for Authors

This is a well-written and comprehensive review of the role of allogeneic stem cell transplantation in patients with Ph+ ALL in the contemporary era. I have only minor observations as follows:

1. Page 2 lines 96-97. I disagree that the prognosis with imatinib+chemotherapy and alloHSCT is considered dismal. With this strategy of treatment, the long-term OS is about 60% (Chalandon Y et al, Blood 2015;125:3711-9 or Candoni Biol Blood Marrow Transplant 2019, or C Junghanss, GMALL EHA 2018) remaining the standard treatment in the first line, outside clinical trials. These references are relevant and should be quoted in the manuscript

2. Page 5, lines 197-205. It is unclear the reason why the lines are in italics.

3. The authors might comment further on their opinion on approaching controversial situations. For example: italics

a. Do they believe CAR-T cells may become an alternative to allogeneic stem cell transplantation in the setting of MRD-positive?

b. Are there patient subgroups for whom a specific treatment modality might be preferred compared to others, e.g. TKI+chemotherapy+alloHSCT for younger and fit patients, but not for older and frailer, or how the therapeutic landscape will change according to some specific genetic features? 

c. What is the possible role of inotuzumab in Ph+ ALL? 

d. In the coming years, with the incorporation of new TKIs and immunotherapies into the earlier phases of treatment, how do the authors suggest approaching the few patients who will relapse? I suggest adding a paragraph for the refractory/relapsed setting

Author Response

Reviewer 1

This is a well-written and comprehensive review of the role of allogeneic stem cell transplantation in patients with Ph+ ALL in the contemporary era. I have only minor observations as follows:

  1. Page 2 lines 96-97. I disagree that the prognosis with imatinib+chemotherapy and alloHSCT is considered dismal. With this strategy of treatment, the long-term OS is about 60% (Chalandon Y et al, Blood 2015;125:3711-9 or Candoni Biol Blood Marrow Transplant 2019, or C Junghanss, GMALL EHA 2018) remaining the standard treatment in the first line, outside clinical trials. These references are relevant and should be quoted in the manuscript

We thank the reviewers for their comment. We have now added ‘Before the discovery of TKIs, Ph+ ALL was considered to have dismal outcomes. With the use of imatinib and allo-HSCT, long-term outcomes saw a significant improvement with 5-year OS rate approaching 50%’ to lines 100-102 on page 3. We have also referenced studies by Chalandon et al. and Candoni et al.

  1. Page 5, lines 197-205. It is unclear the reason why the lines are in italics.

We thank the reviewer for their comment. We have removed the italics from those lines and summarized them in plain language.

  1. The authors might comment further on their opinion on approaching controversial situations. For example: italics
  2. Do they believe CAR-T cells may become an alternative to allogeneic stem cell transplantation in the setting of MRD-positive?

We thank the reviewer for their comment. Currently it is too early (with limited datasets) to comment on whether CAR-T therapy may become an alternative to allogeneic stem cell transplantation in the setting of MRD-positive. We have discussed in lines 325-326 on page 7 that ‘Although promising, these early results need confirmation in larger studies to determine if CAR-T therapy can potentially replace allo-HSCT in high-risk Ph+ ALL.’

  1. Are there patient subgroups for whom a specific treatment modality might be preferred compared to others, e.g. TKI+chemotherapy+alloHSCT for younger and fit patients, but not for older and frailer, or how the therapeutic landscape will change according to some specific genetic features? 

We thank the reviewer for their comment. We have discussed in lines 361-366 on page 8 that ‘In the current era, all transplant eligible patients with Ph+ ALL should have a consultation with a transplant expert to discuss the risks and benefits proceeding with allo-HSCT. Based on the evidence presented in the review, all eligible patients with Ph+ ALL in second remission, those who do not achieve a CMR after three months of frontline therapy and patients harboring high-risk biological anomalies such as IKZF1 should be recommended to undergo allo-HSCT.’

With regards to genetics features, we discuss in lines 107-110 on page 3 that ‘The presence of IKZF1 (plus) abnormalities was also associated with inferior outcomes when treated with ponatinib and hyper-CVAD. Longer follow up of the ponatinib and blinatumomab cohort is needed to see if the combination can overcome the adverse outcomes of IKZF1 (plus) abnormalities.’

In Figure 1, we propose the treatment algorithm for newly diagnosed Ph+ ALL based on the currently available literature.

Lastly, as our review focused on the role of allo-HSCT, we have discussed the findings as well as our opinion in the context of transplant-eligible patients.

  1. What is the possible role of inotuzumab in Ph+ ALL? 
  2. In the coming years, with the incorporation of new TKIs and immunotherapies into the earlier phases of treatment, how do the authors suggest approaching the few patients who will relapse? I suggest adding a paragraph for the refractory/relapsed setting

We thank the reviewer for their comment. We have combined the response to 3c. and 3d. by adding a separate section for relapse and refractory Ph+ ALL and including the role of inotuzumab ozogamicin within that section. The section is between lines 328-346 on page 7 and page 8 and includes the following:

Relapse and Refractory Disease

            The treatment options for R/R ALL include blinatumomab, inotuzumab ozogamicin and CAR-T therapy. For Ph+ ALL, novel TKIs may be added to these agents. The potential benefit of blinatumomab and CAR-T in Ph+ ALL has been extensively discuss be-fore. Studies have shown that inotuzumab ozogamicin, with or without a TKI, is feasible to administer in R/R Ph+ ALL. It can serve as a potential bridge to allo-HSCT in second remission, although more data is needed to understand the potential post-transplant toxicities of using it pre-transplant. The studies so far do not show an in-creased risk of veno-occlusive disease. However, these studies had a small sample size and larger studies are needed to assess the full benefit as well as safety of inotuzumab ozogamicin in both newly diagnosed and R/R Ph+ ALL. Additionally, novel TKIs, such as olverembatinib, have shown encouraging single-agent activity in R/R Ph+ ALL and are now being investigated in various combinations with either chemotherapy or blinatumomab. As these agents are moved to the frontline setting with the hope of inducing deeper and durable remissions, a challenge may arise of treating those patients with disease relapse after these therapies. After thorough testing to characterize the nature of the relapse disease in terms of antigen expression and molecular anomalies, treatment may largely depend on ability to enroll on clinical trials or the utilization of standard of care TKIs, immunotherapy or even conventional chemotherapy.

Reviewer 2 Report

Comments and Suggestions for Authors

 Jamy  and  Badar performed a comprehensive review of the role of alloHSCT in Ph+ALL, that is an evolving matter. The paper is well elaborated, and the information is well balanced. Only minor comments are addressed to authors

1.      The summary and the introduction are practically the same. I suggest modifying the introduction in order to differentiate from the summary

2.      Line 60. I suggest to reference and comment the following paper: Chalandon Y, Rousselot P, Chevret S, Cayuela JM, Kim R, Huguet F, Chevallier P, Graux C, Thiebaut-Bertrand A, Chantepie S, Thomas X, Vincent L, Berthon C, Hicheri Y, Raffoux E, Escoffre-Barbe M, Plantier I, Joris M, Turlure P, Pasquier F, Belhabri A, Guepin GR, Blum S, Gregor M, Lafage-Pochitaloff M, Quessada J, Lhéritier V, Clappier E, Boissel N, Dombret H. Nilotinib with or without cytarabine for Philadelphia-positive acute lymphoblastic leukemia. Blood. 2024 Jun 6;143(23):2363-2372. doi: 10.1182/blood.2023023502. PMID: 38452207.

3.      Line 94-95. Refer to the randomized Phase 3 trial from the GIMEMA Group (ALL2820) ( imatinib+chemotherapy vs Ponatinib+blinatumomab)

4.      LIne 154. I Suggest to reference and comment the  PONALFIL trial, because (different from the ponatinib+HyperCVAD) combines ponatinib and chemotherapy, followed by alloHSCT in almost all patients. Ribera JM, Morgades M, Ribera J, Montesinos P, Cano-Ferri I, Martínez P, Esteve J, Esteban D, García-Fortes M, Alonso N, González-Campos J, Bermúdez A, Torrent A, Genescà E, Maluquer C, Martínez-López J, García-Sanz R. Ponalfil trial for adults with Philadelphia chromosome-positive acute lymphoblastic leukemia: Long-term results. Hemasphere. 2024 Apr 2;8(4):e67. doi: 10.1002/hem3.67. PMID: 38566805; PMCID: PMC10986419.

5.      Iine 165. The updated results of the ponatinib+ blinatumomab have been recently published. Kantarjian H, Short NJ, Haddad FG, Jain N, Huang X, Montalban-Bravo G, Kanagal-Shamanna R, Kadia TM, Daver N, Chien K, Alvarado Y, Garcia-Manero G, Issa GC, Garris R, Nasnas C, Nasr L, Ravandi F, Jabbour E. Results of the Simultaneous Combination of Ponatinib and Blinatumomab in Philadelphia Chromosome-Positive ALL. J Clin Oncol. 2024 Dec 20;42(36):4246-4251. doi: 10.1200/JCO.24.00272. Epub 2024 Jul 19. PMID: 39028925.

6.      Line 165. I suggest including the results of the combination of ponatinib and blinatumomab from the GIMEMA group presented at ASH2024. (Abstract 835)

7.      Lines 197-205. Use the same type of letter as that of the  remaining article

8.      Table 1. Include the results of the data of the GMALL protocol, recently presented at ASH2024 (abstract 961)

9.      Comment briefly on the preliminary data with new TKI inhibitors, such as olverembatinib. Jabbour E, Oehler VG, Koller PB, Jamy O, Lomaia E, Hunter AM, Uspenskaya O, Samarina S, Mukherjee S, Cortes JE, Baer MR, Zherebtsova V, Shuvaev V, Turkina A, Davydkin I, Guo H, Chen Z, Fu T, Jiang L, Wang C, Wang H, Yang D, Zhai Y, Kantarjian H. Olverembatinib After Failure of Tyrosine Kinase Inhibitors, Including Ponatinib or Asciminib: A Phase 1b Randomized Clinical Trial. JAMA Oncol. 2024 Nov 21:e245157. doi: 10.1001/jamaoncol.2024.5157. Epub ahead of print. PMID: 39570620; PMCID: PMC11583018.

Author Response

Reviewer 2

Jamy  and  Badar performed a comprehensive review of the role of alloHSCT in Ph+ALL, that is an evolving matter. The paper is well elaborated, and the information is well balanced. Only minor comments are addressed to authors

  1. The summary and the introduction are practically the same. I suggest modifying the introduction in order to differentiate from the summary

We thank the reviewer for their comment. We have now modified the abstract as well as the introduction to avoid overlapping.

  1. Line 60. I suggest to reference and comment the following paper: Chalandon Y, Rousselot P, Chevret S, Cayuela JM, Kim R, Huguet F, Chevallier P, Graux C, Thiebaut-Bertrand A, Chantepie S, Thomas X, Vincent L, Berthon C, Hicheri Y, Raffoux E, Escoffre-Barbe M, Plantier I, Joris M, Turlure P, Pasquier F, Belhabri A, Guepin GR, Blum S, Gregor M, Lafage-Pochitaloff M, Quessada J, Lhéritier V, Clappier E, Boissel N, Dombret H. Nilotinib with or without cytarabine for Philadelphia-positive acute lymphoblastic leukemia. Blood. 2024 Jun 6;143(23):2363-2372. doi: 10.1182/blood.2023023502. PMID: 38452207.

We thank the reviewer for their comment. We have now referenced the study by Chalandon et al. and discuss in lines 59-63 on page 2 that ‘Another randomized trial of nilotinib, with or without cytarabine during consolidation, showed no difference in major molecular response (71% vs. 77%) or 4-year survival (79% vs. 73%). However, an increased risk of relapse was observed where cytarabine was omitted (31%) vs. not omitted (13%).’

  1. Line 94-95. Refer to the randomized Phase 3 trial from the GIMEMA Group (ALL2820) ( imatinib+chemotherapy vs Ponatinib+blinatumomab)

We thank the reviewer for their comment. We have now added to lines 97-99 on page 2 and 3 that ‘Another ongoing phase III trial (GIMEMA; NCT04722848) is comparing ponatinib-blinatumomab to imatinib-chemotherapy in newly diagnosed Ph+ ALL.’

  1. LIne 154. I Suggest to reference and comment the  PONALFIL trial, because (different from the ponatinib+HyperCVAD) combines ponatinib and chemotherapy, followed by alloHSCT in almost all patients. Ribera JM, Morgades M, Ribera J, Montesinos P, Cano-Ferri I, Martínez P, Esteve J, Esteban D, García-Fortes M, Alonso N, González-Campos J, Bermúdez A, Torrent A, Genescà E, Maluquer C, Martínez-López J, García-Sanz R. Ponalfil trial for adults with Philadelphia chromosome-positive acute lymphoblastic leukemia: Long-term results. Hemasphere. 2024 Apr 2;8(4):e67. doi: 10.1002/hem3.67. PMID: 38566805; PMCID: PMC10986419.

We thank the reviewer for their comment. We have now discussed the PONALFIL trial in lines 158-162 on page 4. ‘In the PONALFIL trial, ponatinib was combined with daunorubicin, vincristine and prednisone with 26 patients proceeding to allo-HSCT. Majority of the patients received MAC and the 3-year event-free survival (EFS) and OS for these patients was 70% and 96%, respectively. One patient died from TRM in 3 years.’

  1. Iine 165. The updated results of the ponatinib+ blinatumomab have been recently published. Kantarjian H, Short NJ, Haddad FG, Jain N, Huang X, Montalban-Bravo G, Kanagal-Shamanna R, Kadia TM, Daver N, Chien K, Alvarado Y, Garcia-Manero G, Issa GC, Garris R, Nasnas C, Nasr L, Ravandi F, Jabbour E. Results of the Simultaneous Combination of Ponatinib and Blinatumomab in Philadelphia Chromosome-Positive ALL. J Clin Oncol. 2024 Dec 20;42(36):4246-4251. doi: 10.1200/JCO.24.00272. Epub 2024 Jul 19. PMID: 39028925.

We thank the reviewer for their comment. We have now discussed the updated trial results in lines 169-172 on page 4. ‘In the study of ponatinib plus blinatumomab, only two out of 60 patients proceeded to allo-HSCT in first remission. With a median follow-up of 24 months, the 3-year EFS and OS were 77% and 91%, respectively. Seven patients relapsed (2=systemic relapse, 4=isolated CNS relapse, 1=extramedullary Ph negative ALL relapse).’

  1. Line 165. I suggest including the results of the combination of ponatinib and blinatumomab from the GIMEMA group presented at ASH2024. (Abstract 835)

We thank the reviewer for their comment. We have now discussed the GIMEMA ALL2820 results in lines 173-177 on page 4. ‘In the intermediate analysis of the GIMEMA ALL2820 trial, 133 patients received ponatinib plus blinatumomab. The decision to proceed to allo-HSCT was based on IKZF1 (plus) at diagnosis and/or MRD persistence. To date, 12% of the patients have proceeded to transplant and with a median follow-up of 6.4 months, the estimated 18-month OS is 91.6%.’

  1. Lines 197-205. Use the same type of letter as that of the  remaining article

We thank the reviewer for their comment. We have removed the italics from those lines and summarized them in plain language.

  1. Table 1. Include the results of the data of the GMALL protocol, recently presented at ASH2024 (abstract 961)

We thank the reviewer for their comment. Unfortunately, abstract 961 from ASH 2024 does not provide separate results for Ph+ ALL. They provide results for all B-cell ALL types as well as T-cell ALL. As our table is only focused on Ph+ ALL results, we are unable to add those specific results of the GMALL trial to our table at this point.

  1. Comment briefly on the preliminary data with new TKI inhibitors, such as olverembatinib. Jabbour E, Oehler VG, Koller PB, Jamy O, Lomaia E, Hunter AM, Uspenskaya O, Samarina S, Mukherjee S, Cortes JE, Baer MR, Zherebtsova V, Shuvaev V, Turkina A, Davydkin I, Guo H, Chen Z, Fu T, Jiang L, Wang C, Wang H, Yang D, Zhai Y, Kantarjian H. Olverembatinib After Failure of Tyrosine Kinase Inhibitors, Including Ponatinib or Asciminib: A Phase 1b Randomized Clinical Trial. JAMA Oncol. 2024 Nov 21:e245157. doi: 10.1001/jamaoncol.2024.5157. Epub ahead of print. PMID: 39570620; PMCID: PMC11583018.

We thank the reviewer for their comment. We have now discussed in lines 338-340 on page 7 that ‘Additionally, novel TKIs, such as olverembatinib, have shown encouraging single-agent activity in R/R Ph+ ALL and are now being investigated in various combinations with either chemotherapy or blinatumomab.’